# Research on the General Failure Law of a CTRC Column by Modeling FEM Output Data

**DOI:** 10.3390/ma15176058

**Published:** 2022-09-01

**Authors:** Zijie Shen, Bai Liu, Guangchun Zhou

**Affiliations:** Key Lab of Smart Prevention and Mitigation of Civil Engineering Disasters of the Ministry of Industry and Information Technology, School of Civil Engineering, Harbin Institute of Technology, Harbin 150090, China

**Keywords:** CTRC column, finite element model, eccentric loading, correlation method, stressing state pair, mutation, law

## Abstract

In this paper, a finite element model (FEM) is developed based on a set of circular steel tube reinforced concrete (CTRC) columns with axial compression and eccentric compression tests. The stressing state characteristics of the FEM are modeled in the form of characteristic pairs (mode-characteristic parameters) based on the structural stressing state theory and the proposed correlation modeling method. The slope increasing criterion is applied to the correlation characteristic parameter curve to obtain the characteristic point Q where the CTRC stressing state undergoes a qualitative change, and the characteristic point Q is defined as the new failure load point of the CTRC column. By selecting the element strain energy density at different locations of the FEM for correlation stressing state modeling and dividing the correlation stressing state sub-modes (concrete, steel tube, vertical reinforcement, and stirrup reinforcement), the structural stressing state theory and the rationality of the proposed correlation stressing state modeling method are verified. In addition, the certainty and reasonableness of the failure load points of the CTRC columns are revealed and verified.

## 1. Introduction

A concrete-filled steel tube is a combined structure developed by combining the advantages of steel and concrete. Since the 1960s, concrete-filled steel tube structures have received attention in developed countries such as the United States, European countries, and Japan and have been widely used in engineering. Many engineering applications of concrete-filled steel tubes have also promoted research on concrete-filled steel tube structures. The worldwide research on the mechanical behavior of concrete-filled steel tube columns is divided into the axial and the eccentric bearing capacity, and the classical paradigm, as shown in Figure 1, has been gradually formed during the research.

Table 1 lists the CTRC axial bearing capacity expressions based on the ultimate state, the plastic state, and the stable bearing capacity of CTRC columns and summarizes their limitations.

Summarizing the expressions in Table 1, the formula for the bearing capacity of short columns is often based on the ultimate state of concrete or steel tube, and then the expression for the bearing capacity of steel tube concrete column is derived according to the ultimate state. For long columns, more consideration is given to the stable bearing capacity of the member. In addition to the correction based on the Eulerian restraining force, the actual engineering or design codes often obtain the empirical stability coefficient by fitting the stable bearing capacity of the actual test. The limitations of the expressions in Table 1 are considered comprehensively as follows.

(1)The choice of the k value is based on the researcher’s experience, generally *k* = 4.(2)The hardening of steel is considered according to the ultimate state, and there are limitations. The expression uses the concrete strength enhancement factor multiplied by the load-bearing capacity of the steel tube, so the mechanics are unclear.(3)The moment when the vertical stress of the steel tube reaches the uniaxial yield point is defined as the moment when the strength of the steel tube changes from elastic to plastic, which is inconsistent with the actual situation.(4)The load-bearing capacity calculation process ignores the interaction between steel tubes and concrete.

Table 2 lists typical calculation theories and expressions for the eccentric bearing capacity of concrete-filled steel tubes. In general, the research on the eccentric bearing capacity of concrete-filled steel tubes is less than the axial bearing capacity. However, from the standpoint of the research, the eccentric bearing capacity of concrete-filled steel tube columns is also based on the internal force balance condition of the ultimate state or the empirical formula obtained by fitting the test results of the eccentric load stability bearing capacity. The more classic expressions for calculating the eccentric bearing capacity of these concrete-filled steel tube columns are often complicated to properly consider the impact of different load eccentricities on the bearing capacity, and there is no unified foothold. In actual engineering, the design of concrete-filled steel tube compression-bending members often further reduces the load-bearing capacity for a safer and conservative design.

(1)The expression is derived from the internal force equilibrium condition, which needs to be adjusted manually when calculating the bearing capacity of a small eccentric.(2)The relevant equation of the linear expression does not consider the plastic development of the cross-section.(3)The calculation results have the same effect on various slenderness ratio members with different load eccentricities, which is inconsistent with the actual situation.

Summarizing the features of the concrete-filled steel tube (CFST)’s bearing capacity calculation formulas of concrete-filled steel tube columns under axial compression and eccentric compression based on the classic theory, it can be found that there are several common problems.

First, the derivation of the bearing capacity expression is often based on the ultimate state of the concrete-filled steel tube column or the fleeting moment of plasticity, rather than focusing on the evolution law of the failure process of the concrete-filled steel tube column.

Second, the determination of the ultimate state of steel tube or concrete is often based on the longitudinal strain value or the peak value of the force−displacement curve at specific points, and the strain or strain energy data obtained by an experiment or simulation are not fully utilized.

Third, the final form of the bearing capacity expression is often based on the linear superposition of the bearing capacity provided by the various parts of the CFST column and does not consider the interaction between the various components/elements.

Fourth, classical structural analysis theory considers uncertainty in the ultimate state as an inherent property of structural load-carrying capacity. However, at the same time, engineers and researchers are constantly pursuing more accurate structural designs to overcome the effects of uncertainty. Therefore, the problem of the structural load carrying capacity has always been an open problem.

The above four problems are a classical proposition in structural engineering. Furthermore, researchers believe that the ultimate state of the load-carrying capacity of a structure has inherent uncertainty. In recent years, Zhou proposed the structural stressing state theory to solve this problem and considered that structural damage is a process with a starting and end point. Moreover, structural stressing state modeling analysis has revealed its deterministic failure starting point and an elasto-plastic branching point in more than a dozen different working conditions and structures. These include steel nodes [9], reinforced masonry shear walls [10,11], steel frames [12], stainless steel tubular concrete columns [13], spiral reinforcement concrete columns [14], steel tube concrete arches [15,16], steel tube concrete arch bridges [17], continuous curved box girder bridges [18], and space mesh shells [19]. Other structures under static conditions, hysteresis conditions, and simulated shaking table conditions have revealed the deterministic failure starting points and elastoplastic branch points. The general damage law of the structural damage process has been verified and revealed.

The authors of this paper developed a finite element model (FEM) based on the axial and eccentric compression tests of 12 CTRC columns in the literature [20]. Based on the structural stressing state theory and the correlation stressing state modeling method, the failure starting point of CTRC columns under axial and eccentric loads is revealed from several perspectives. The modes and characteristic parameters obtained by multiple correlation modeling methods all have noticeable mutation near the failure starting point of the CTRC column. Furthermore, in the vicinity of the characteristic point, the nephogram of the FEM also shows a pronounced mutation feature. This reveals the general failure law of CTRC columns under axial and bias conditions and that it is possible to achieve the optimal design of a CTRC column by using the failure point of the CTRC column as the control point of the structural design.

## 2. Outline of Structural Stressing State Theory

### 2.1. Stressing State Theory

The structural stressing state theory considers that the damage of a structure is a process with a starting point and an end point. The ultimate state of the structure is uncertain, but the starting point of failure is deterministic. The modeling of the stressing state with the data of strain, displacement, and strain energy during the loading of the structure is used to characterize the stressing state of the structure and the evolution of the damage process. The starting point of the structural damage process can be obtained by determining the moment of qualitative change in the evolution process, in which the load corresponding to the starting point of failure is defined as the failure load.

### 2.2. Correlation Modeling Methods

Characteristic pairs refer to characteristic vectors and characteristic parameters. This concept originates from linear algebra and is widely used in computer vision and machine learning. The characteristic vector (matrix) describes the state of the structure at each moment of the loading process, while the characteristic parameter preserves the evolution trend of the stressing state of the structure. By judging the mutation in the evolution of the characteristic parameters, the characteristic points of the stressing state of the structure can be obtained. The correlation modeling method describes the stressing state of the structure and its evolution law by modeling the degree of correlation between the elements of the structure or finite element model. The characteristic pairs established based on the correlation method are called correlation characteristic pairs. The correlation modeling method proposed in this paper for modeling CTRC column finite element models is as follows:

Step 1: Four elements are selected on the finite element model (concrete, vertical reinforcement, steel tube, and Stirrup reinforcement) to output the elastic element strain energy density during the whole loading process. Then, the strain energy density is normalized for the whole loading process. The vector shown in Equation (1) is obtained.
(1)$$S_{{}_{j,\mathrm{norm}}}^{\mathrm{ele}}={\left[e_{1}\;e_{2}\;e_{3}\;e_{4}\right]}_{j,\mathrm{norm}}$$
where *e* denotes the normalized elastic element strain energy density of the selected element in the finite element model, and the subscript denotes the number of units. *j* denotes the ordinal number of the load step. The symbols in the following equations have the same meanings.

Step 2: Take the difference between the two normalized elastic element strain energy densities in Equation (1) and perform the integration operation. Then take the absolute value of the integrated value and normalize it to obtain the vector shown in Equation (2). Equation (2) is then simplified to obtain Equation (3).
(2)$$S_{j,\mathrm{norm}}^{\mathrm{Difference}}={\left[\left|\int_{0}^{F}e_{1}-e_{2}dF\right|\ldots \left|\int_{0}^{F}e_{3}-e_{4}dF\right|\right]}_{j,\mathrm{norm}}^{6}$$
(3)$$S_{j,\mathrm{norm}}^{\mathrm{Difference}}={\left[D_{1–2}\ldots D_{3–4}\right]}_{j,\mathrm{norm}}^{6}={\left[D_{a}\ldots D_{f}\right]}_{j,\mathrm{norm}}^{6}$$
where *F* denotes the load step in the loading process and *D* denotes Equation (2), with the simplified representation of each element in the vector. The superscript 6 indicates the number of elements in the vector.

Step 3: The elements normalized in Equation (3) are two-by-two differenced and integrated, and once again, the integrated values are taken as absolute values and normalized to obtain the vector shown in Equation (4), which is the correlation mode of the CTRC column finite element model.
(4)$$R_{j,\mathrm{norm}}^{}={\left[\left|\int_{0}^{F}D_{a}-D_{b}dF\right|\ldots \left|\int_{0}^{F}D_{e}-D_{f}dF\right|\right]}_{j,\mathrm{norm}}^{15}$$
(5)$$R_{j,\mathrm{norm}}^{}={\left(\sum_{N=1}^{15}\left|\int_{0}^{F}D_{a}-D_{b}dF\right|\right)}_{j,\mathrm{norm}}$$
where the superscript 15 indicates the number of elements in the vector.

In this paper, the finite element model of a CTRC column is divided into four parts: concrete, steel tube, vertical reinforcement, and stirrup reinforcement. Then, three different element selection methods are carried out to model the correlation stressing states for each of the four parts, i.e., three correlation modes are obtained. The details are elaborated in Section 4 of this paper.

### 2.3. Slope Increment Criterion

The slope increment criterion [19] is a mutation criterion proposed by the structural stressing state theory. Using the slope increment criterion, the mutation in the failure evolution process of the structure can be determined. The position where the slope increment of the characteristic parameter curve has a mutation is defined as when the structural stressing state has a qualitative change. The expression of the slope increment criterion is shown in Equation (6).
(6)$$S_{j}-S_{j-1}\geq W$$

Among them, S represents the slope of the characteristic parameter curve, j represents the load step, and W is the ultimate determined based on experience.

## 3. Brief of Test/FEM

### 3.1. Test Overview

In this paper, a finite element model was developed based on the tests of 12 axial and deflected CTRC columns in the literature [20]. The structural parameters of the 12 CTRC columns are shown in Table 3, where D denotes the diameter of the column, t denotes the thickness of the steel tube, L denotes the length of the column, and λ denotes the length-to-thin ratio of the column. The naming rules of the specimens are as follows: in CTRC-200-6-25, for example, 200 indicates that the diameter of the CTRC column is 200 mm, 6 indicates the length/diameter ratio of the CTRC column is 6, and 25 indicates the eccentricity of the eccentric loading is 25 mm.

The loading device of the CTRC column is shown in Figure 2. The axial load is transferred to the CTRC column through the hydraulic loading device with the head hinge support, and the eccentricity of the axial loading is controlled by the distance between the head hinge support and the symmetry axis of the section.

### 3.2. Finite Element Model

The finite element model established in this paper was based on the experiment of [20], and the finite element model of the 12 CTRC columns in Table 3 was established using ABAQUS software (Abaqus/CAE 2020, SIMULIA, Providence, RI, USA). The test used C50 concrete as the core concrete and 1.5 mm thick Q235 steel to make round steel tubes. The vertical bar was an HRB400 steel bar with a diameter of 20 mm, measured *f**_y_* = 477.2 MPa, and measured *f**_u_* = 647.88 MPa, and the stirrup was an HRB300 steel bar with a diameter of 8 mm and a yield strength standard value of 300 MPa. Figure 3 shows the meshing of each part of the FEM. The ends of the CTRC column were artificially simplified to end plates with an elastic modulus 1000 times that of steel, and Poisson’s ratio γ = 0.1. The concrete element was C3D8R, the steel tube element was S4R, and the steel reinforcement element was T3D2. The end slab was in restrained contact with the concrete, and the steel reinforcement was embedded in the concrete. The steel tube was in normal hard contact with the concrete and the friction coefficient was set to 0.6.

The numerical simulation of the confined concrete was based on ABAQUS’s concrete damage plasticity and adopted the modified Mander model. As the constrained concrete constitutive, the Q235 steel adopted the five-fold line constitutive model of steel, and the HRB400 and HRB300 steel bars were based on GB/50010 (2015) [21], adopting a double-fold line constitutive model, elastic modulus *E*_1_ = 2.0 × 10^5^ Mpa, and strengthening modulus *E*_2_ = 0.005*E*_1_. The simplification of the constrained concrete tension was set to a linear relationship. The expressions of the confined concrete and steel constitutive models are shown in Table 4.

Among them, the parameters in the expressions of the constitutive model of the steel tube and confined concrete were selected as shown in Table 5.

As shown in Figure 4, the finite element model was compared with the force−displacement curves of the three eccentric CTRC_200_6 columns in [20]. It can be concluded that the choices of various parameters of the finite element model established in this study are reasonable.

## 4. Stressing State Analysis

### 4.1. Overall Characteristic Pairs

Taking the finite element model CTRC_200_6_25 as an example, the element elastic strain energy density (ELSED) of the finite element concrete, steel tube, vertical reinforcement, and stirrup reinforcement was modeled according to the process in Section 2.2 of this paper. Figure 5b shows the characteristic parameter curves of the four parts of the CTRC column, which were summed and normalized to obtain the correlation characteristic parameter curves characterizing the overall stressing state of the CTRC column shown in Figure 5a. When the axial force reached 969 kN, the stressing state of the CTRC column changed abruptly. The characteristic point is defined as the damage starting point of the CTRC column under eccentric load conditions, and the load corresponding to the damage starting point is defined as the damage load point. Figure 5b shows that all four parts of the CTRC column show different degrees of mutations, indicating that the four parts of the CTRC column had very high coordination during the loading process. The characteristic point Q as the failure starting point of the CTRC column had a very high reference value for the design of the CTRC column.

Figure 6 is a nephogram of the concrete damage plasticity development of the longitudinal section of the concrete component of CTRC_200_6_25 near the failure load point. It can be seen in the figure that the concrete damage plasticity (PEEQ) of the longitudinal section of the CTRC_200_6_25 concrete column developed from the initial scattered distribution to the development mode of the concrete damage plasticity near the failure starting point. The other eccentrically loaded CTRC columns also had similar nephogram characteristics.

In the following, the correlation stressing state characteristic pairs characterizing the concrete, steel tube, stirrup, and vertical reinforcement were obtained by modeling and analyzing the correlation stressing state of each part of the CTRC column. The mutations in the stressing state near the characteristic points were verified.

### 4.2. Part Characteristic Pair

The correlation stressing state modeling analysis of the steel tube is as follows: The strain energy density (ELSED) of the steel tube element was the output, and the correlation numerical mode and characteristic parameters were established according to the process of the correlation modeling analysis method described in Section 2.2. In order to verify the rationality of the correlation modeling analysis method and the certainty and uniformity of the mutations in the correlation stressing state near the characteristic point Q, the mutations in the stressing state of the steel tube near the characteristic point Q are revealed by three element selection schemes. As shown in Figure 7, the correlation modes 1, 2, and 3 all bifurcated or turned around 969 kN. The characteristic parameter curves of the correlation stressing state and the slope increment curves corresponding to modes 1, 2, and 3 also show bifurcation near the characteristic point, and the mutation of the slope increment curve is especially obvious near the characteristic point.

For CTRC columns, the most critical load-bearing part is the core concrete. The three element selection methods and the correlation state modeling process described in Section 2.2 were used to obtain the three correlation state modes 1, 2, and 3, as shown in Figure 8. The correlation state modes and characteristic parameters are the same as those of the steel tube part, and all three modes show very obvious bifurcations and transitions near the characteristic points. Among them, the curve of mode 2 shows an undeniable clustering phenomenon near the characteristic point, and the corresponding correlation characteristic parameter also shows an apparent sharp point near the characteristic point.

The correlation between the stressing state modes and the characteristic parameters obtained by different element selection methods show significant mutations in the vicinity of the same characteristic point. This further verifies the rationality of the correlation modeling method and the stability of the revealed characteristic points. Moreover, this phenomenon also indicates that the CTRC column had good synergistic working performance between the steel tube and the core concrete during the loading process. When the CTRC column, as a whole, entered the failure load, the steel tube and the core concrete also started to enter the failure state.

The following correlation modeling analysis method described in Section 2.2 was used to characterize and reveal the mutation characteristics of the CTRC column during the damage process. Figure 9a–c shows the three correlation modes characterizing the stressing states of the stirrup reinforcement of the finite element model CTRC_200_6_25. Figure 9d–f shows the three correlation characteristic modes characterizing the stressing state of the vertical reinforcement of the finite element model CTRC_200_6_25.

As can be seen in the figure, the evolution of the stressing state of the CTRC column stirrup and vertical reinforcement characterized by the correlation modeling analysis method is equally evident in the bifurcation and turning characteristics near its characteristic points. Moreover, this failure law is not a special case of the finite element model CTRC_200_6_25. All of the 12 CTRC columns modeled in this paper exhibit such a failure law during the damage process.

## 5. Conclusions

The authors of this paper developed a finite element model based on the 12 CTRC columns in [20]. Then, the output elastic element strain energy density (ELSED) was modeled as the correlation stressing state mode, with the characteristic parameters based on the structural stressing state theory and the correlation modeling analysis method. The slope increment criterion was used to determine the characteristic points in the evolution of the correlation characteristic parameter curves, and they were defined as the starting points of the damage process. At the same time, the corresponding load was defined as the damage load. In addition, the finite element model of the CTRC column was divided according to the core concrete, steel tube, vertical reinforcement, and stirrup reinforcement and determined the correlation modes of each part of the CTRC column with the mutation characteristics of the characteristic parameters near the characteristic points. The main findings of this paper are as follows.

(1)The applicability of the structural stressing state theory to CTRC columns was verified. A correlation modeling analysis method was proposed to establish the stressing state mode and characteristic parameters of CTRC columns to characterize the evolution of their stressing states. In addition, the location of the characteristic point was determined by the slope increment criterion, and the rationality of the modeling method was verified by the mutation of the correlation model and characteristic parameters near the characteristic point.(2)The damage starting point of the CTRC column had a very stable value. The mode and characteristic parameters obtained by the correlation modeling analysis of the finite element model of the CTRC column well characterized the evolution of the stressing state of the CTRC column. Moreover, applying the slope increment criterion to the correlation characteristic parameter curves revealed the location of the characteristic points.(3)The finite element model of the CTRC column was divided according to the core concrete, steel tube, vertical reinforcement, and stirrup reinforcement. The correlation stressing state modes of each part of the CTRC column obtained by different element selection methods can be a very obvious bifurcation or turning phenomenon near the characteristic points. Moreover, the slope increment curves of the correlation characteristic parameter curves show the mutation characteristic near the characteristic point especially.

## Figures and Tables

**Figure 1 materials-15-06058-f001:**
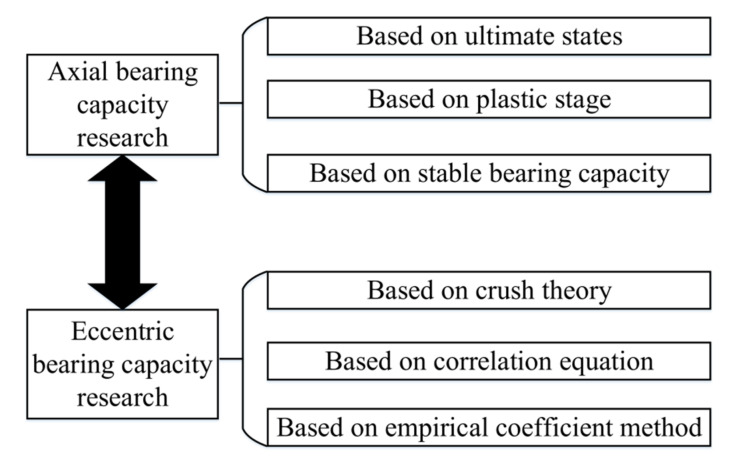
Classical concrete-filled steel tube research system and paradigm.

**Figure 2 materials-15-06058-f002:**
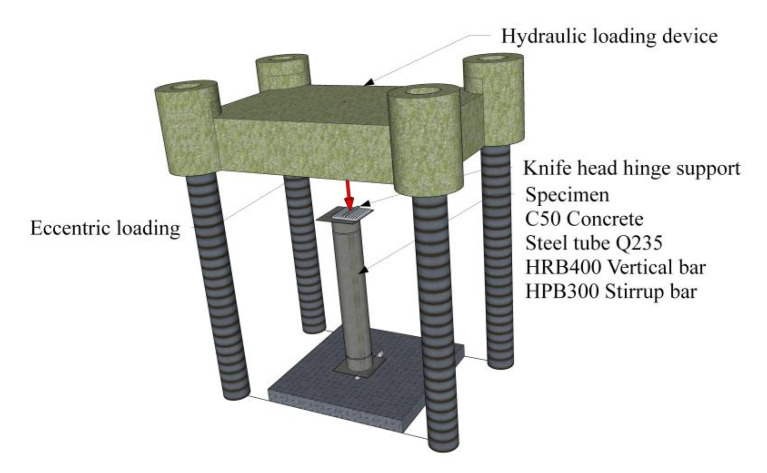
Test device diagram.

**Figure 3 materials-15-06058-f003:**
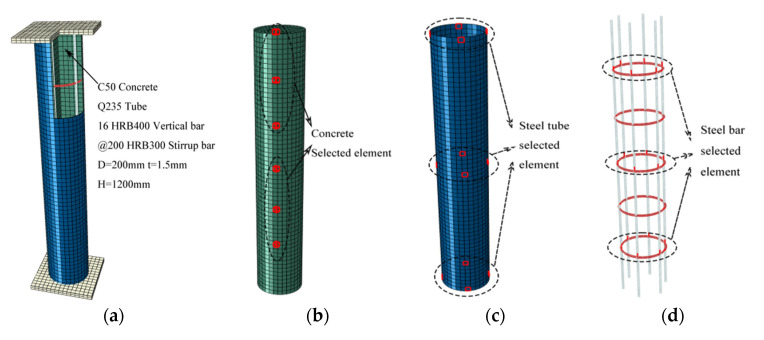
Finite element model of CTRC. (**a**) CTRC_200_6; (**b**) concrete element selection; (**c**) steel tube element selection; (**d**) steel bar element selection.

**Figure 4 materials-15-06058-f004:**
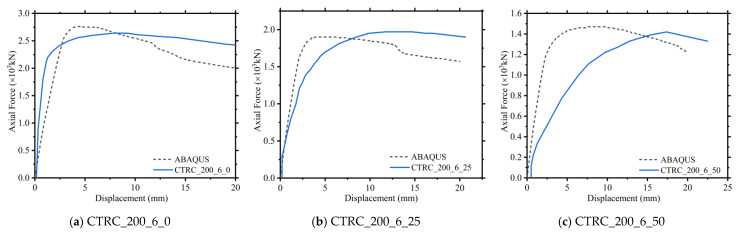
Axial force−displacement curves.

**Figure 5 materials-15-06058-f005:**
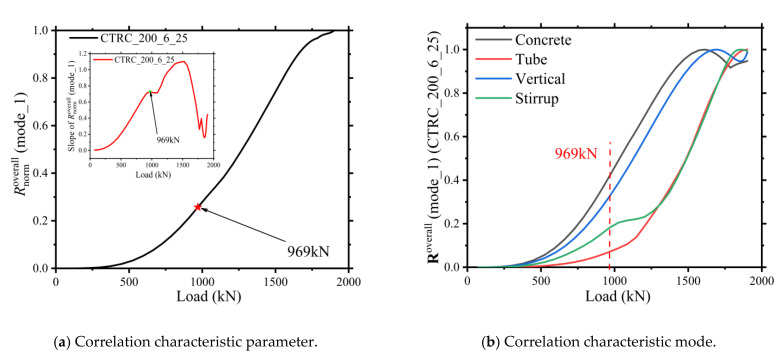
Overall characteristic pairs of CTRC_200_6_25.

**Figure 6 materials-15-06058-f006:**
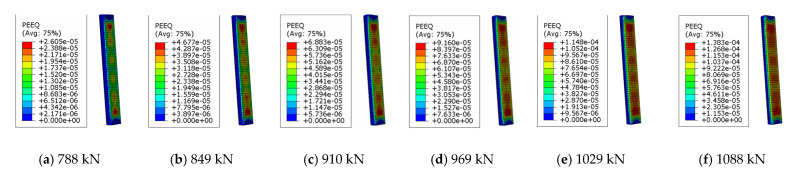
QEET nephograms of CTRC_200_6_25.

**Figure 7 materials-15-06058-f007:**
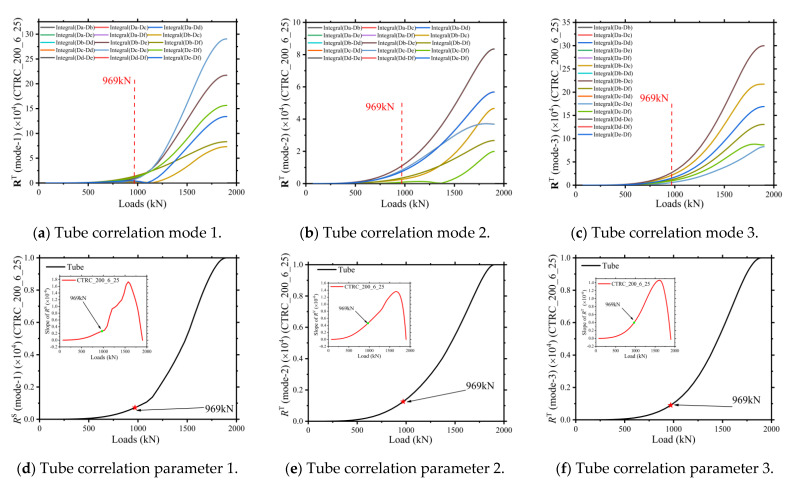
Tube characteristic pairs of CTRC_200_6_25.

**Figure 8 materials-15-06058-f008:**
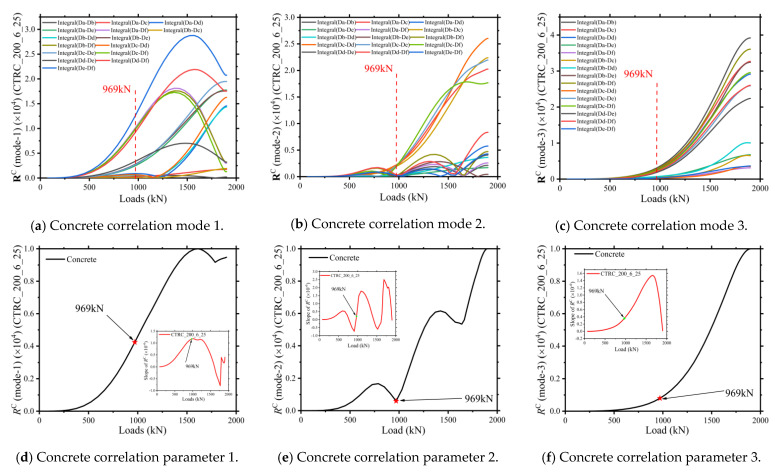
Concrete characteristic pairs of CTRC_200_6_25.

**Figure 9 materials-15-06058-f009:**
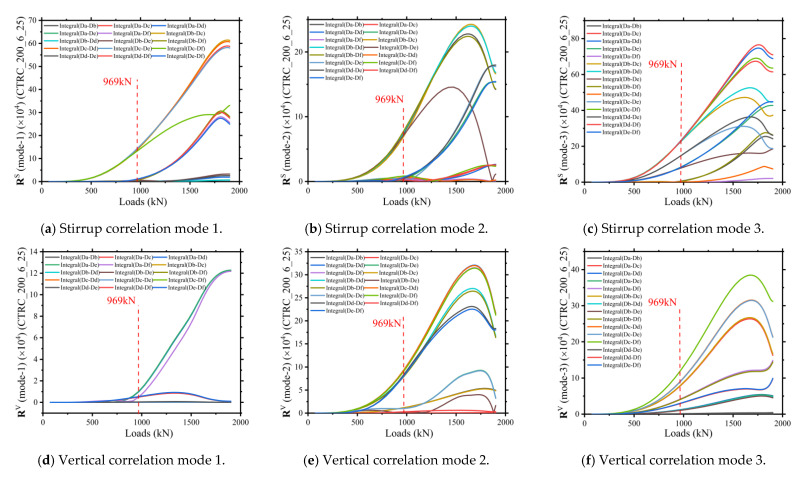
Stirrup bar characteristic pairs of CTRC_200_6_25.

**Table 1 materials-15-06058-t001:** Ultimate axial loading capacity.

Researchers	Foothold	Expression
Knowles, B. et al. [1]	Ultimate states	$N_{u}=f_{ck}A_{c}+kf_{y}A_{s}$
Sen, H.K. [2]		$N_{u}=f_{ck}A_{c}+\sigma_{st}A_{s}+kf_{y}A_{s}$
Neogi, P.K. [3]	Plasticity stage	$N_{u}=f_{y}A_{s}+f_{ck}A_{c}\sqrt{\frac{f_{y}}{0.0018E_{s}}}$
Cai, S.H. [4,5]	Stable bearing capacity	$\varphi_{l}=1-0.115\sqrt{L/D-4}$

where *N_u_* denotes the ultimate bearing capacity, *f_ck_* denotes the restrained concrete strength, *A_c_* denotes the core concrete cross-sectional area, *As* denotes the steel cross-sectional area, *f_y_* denotes the steel yield strength, *k* is the coefficient, and *E_s_* is the steel modulus of elasticity.

**Table 2 materials-15-06058-t002:** Eccentric stability bearing capacity.

Researchers	Foothold	Expression
Zhong Shantong [6]	Crush theory	$N\leq \varphi_{e}N_{0}$ $\varphi_{e}=N_{cr,e}/N_{0}=f\left(f,f,\alpha ,\lambda ,e/r\right)$
W. Furlong [7]	Correlation equation	$\frac{N}{N_{cr}}+\frac{M_{eq}}{\left(1-N/N_{e}\right)M_{u}}=1$
W. Furlong [8]	Empirical coefficient method	$N\leq \varphi_{l}\varphi_{e}N_{0}$ $\varphi_{e}=\frac{1}{1+1.85e_{0}/r_{c}}\;(e_{0}/r_{c}\leq 1.55)$ $\varphi_{e}=\frac{0.4}{e_{0}/r_{c}}\;(e_{0}/r_{c}>1.55)$

where *N_cr_* is the critical bearing capacity and $\varphi_{e}$ is the stability coefficient. The limitations of the expressions in Table 2 are considered comprehensively as follows.

**Table 3 materials-15-06058-t003:** Specimen configuration [20].

Specimen Number	D (mm)	t (mm)	L (mm)	λ (Slenderness Ratio)
CTRC-200-6-0	200	1.5	1200	24
CTRC-200-6-25	200	1.5	1200	24
CTRC-200-6-50	200	1.5	1200	24
CTRC-240-6-0	240	1.5	1440	24
CTRC-240-6-25	240	1.5	1440	24
CTRC-240-6-50	240	1.5	1440	24
CTRC-200-10-0	200	1.5	2000	40
CTRC-200-10-25	200	1.5	2000	40
CTRC-200-10-50	200	1.5	2000	40
CTRC-240-10-0	240	1.5	2400	40
CTRC-240-10-25	240	1.5	2400	40
CTRC-240-10-50	240	1.5	2400	40

**Table 4 materials-15-06058-t004:** Constitutive model of FEM.

Steel Tube	Confined Concrete
$\sigma_{s}=\{\begin{array}{ll} E_{s}\varepsilon_{s} & \varepsilon_{s}\leq \varepsilon_{p}\\ -A\varepsilon_{s}^{2}+B\varepsilon_{s}+C & \varepsilon_{p}<\varepsilon_{s}\leq \varepsilon_{y1}\\ f_{y} & \varepsilon_{y1}<\varepsilon_{s}\leq \varepsilon_{y2}\\ f_{y}\left[1+0.6(\varepsilon_{s}-\varepsilon_{y2})/(\varepsilon_{u}-\varepsilon_{y2})\right] & \varepsilon_{y2}<\varepsilon_{s}\leq \varepsilon_{u}\\ 1.6f_{y} & \varepsilon_{s}>\varepsilon_{u} \end{array}$	$\sigma =\{\begin{array}{ll} E_{c}\varepsilon +\frac{\left(f_{e}-E_{c}\varepsilon_{e}\right)}{\varepsilon_{e}^{2}}\varepsilon^{2} & 0\leq \varepsilon \leq \varepsilon_{e}\\ f_{c}-\frac{\left(f_{c}-f_{e}\right)}{{\left(\varepsilon_{0}-\varepsilon_{e}\right)}^{2}}\times (\varepsilon -\varepsilon_{0}) & \varepsilon_{e}\leq \varepsilon \leq \varepsilon_{0}\\ f_{c}-\beta \frac{f_{c}}{\varepsilon_{0}}\times (\varepsilon -\varepsilon_{0}) & \varepsilon >\varepsilon_{0} \end{array}$

**Table 5 materials-15-06058-t005:** Constitutive model parameter selection.

Steel Tube	Confined Concrete
$\varepsilon_{p}=0.8\times \frac{f_{y}}{E_{s}}$ $\varepsilon_{1}=1.5\varepsilon_{p}$ $\varepsilon_{y2}=10\varepsilon_{p}$ $\varepsilon_{u}=100\varepsilon_{p}$ $\;A=0.2f_{y}/{\left(\varepsilon_{y1}-\varepsilon_{p}\right)}^{2}$ $B=2A\varepsilon_{y1}$ $C=0.8f_{y}+A\varepsilon_{p}^{2}-B\varepsilon_{p}$ $\;f_{y}=364.3\;\mathrm{MPa}\;$ $f_{u}=449\;\mathrm{MPa}$	$f_{c}=40.29\;\mathrm{MPa}$ $f_{e}=\frac{f_{c}\times f_{c}}{f_{cc}}$ $f_{cc}=f_{c}\left[-0.413+1.413\sqrt{1+11.4\frac{f_{l}}{f_{c}}}-2\frac{f_{l}}{f_{c}}\right]$ $E_{c}=4730\sqrt{f_{c}}$ $\varepsilon_{e}=\varepsilon_{c}=\left(700+172\sqrt{f_{c}}\right)\times 10^{-6}$ $\;\varepsilon_{o}=\varepsilon_{cc}-\left(\frac{f_{cc}}{f_{c}}-1\right)\varepsilon_{c}$ $\varepsilon_{cc}=\left[1+5\left(\frac{f_{cc}}{f_{c}}-1\right)\right]\varepsilon_{c}$ $f_{t}=0.26f_{cu}^{2/3}$ $\varepsilon_{t}=65\times 10^{-6}f_{t}^{0.54}$ $\varepsilon_{tu}=25\varepsilon_{t}$ $f_{tu}=0.02f_{t}$ β=0.2

## Data Availability

The data presented in this study are available on request from the corresponding author.

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
