# Peer review of "Research on the General Failure Law of a CTRC Column by Modeling FEM Output Data"

_materials, 2022, doi:10.3390/ma15176058_

Round 1

Reviewer 1 Report

1.      The title clearly reflects the content of the study.

2.      The abstract includes the purpose and methodology.

3.      The language and expression in the manuscript is sufficient.

4.      The introduction section is considered well and explaining of the problem and objectives of study is satisfactory.

5.      The methodology and applications are sufficient and the purpose of the study is well-defined.

6.      FEM models well defined and results are satisfactory.

7.      The results are discussed according to the purpose. Discussions are quite discussed with together results and are satisfactory.

But some points need to be reconsidered and some corrections.

1.      Last sentence of the abstract should be rewritten.

2.      CFST should be written in open form initially, and the abbreviation is to follow next in parenthesis (line 78) 

3.      It is complicated from 106 to 114, make these lines clear.

4.      Author should make it clear that, how a best design of a CFST is achieved, for instance the question that whether the concrete firstly fails or steel tube, must be answered in the introduction part. If it is stated in the text, it can be emphasized in the introduction section initially.

5.      The reason of the inherent property of uncertainty must be declared/explained in details in introduction part (lines 93-94). It is accepted so, but why, still the question is open.

6.      The behavior of each part is shown in graphics very well. But results, output must be stated clearly in the conclusion part point by point for each part’s plot. The conclusion should be developed more.

Reviewer 2 Report

This paper dealt with the finite element analysis of reinforced circular steel tubes. Specifically, the work is about the axial bearing capacity of concrete-filled tubes. Many practical applications exist for the problem tackled in this paper.

1.      The abstract needs to be re-written to indicate the major observations from the study

2.      Overall, the presentation of the work needs to be thoroughly improved. For instance:

a.      Introduction

                                                    i.     Surprisingly, the citations/references are missing in the introduction. Many of the statements in the introduction need to be supported by citations.

                                                   ii.     The source of the expression in Table 1 should be provided.

                                                  iii.     Many of the parameters in the equations contained in Table 1 are not defined. It will be good to correct this. It is better to provide the limitations listed in Table 1 as a separate paragraph.

                                                  iv.     The paragraph between Table 1 and Table 2 should be re-formatted.

                                                   v.     There are many instances of “Reference source not found” within the manuscript. The authors should do a thorough check and correct all of them.

                                                  vi.     Overall, the introduction should be improved to give an account of related studies that have done investigations on similar problems. This will give an idea about the gap in the literature. In the current form of the manuscript, the gap in the literature that leads to the research reported in this manuscript is not clear at all.

                                                vii.     The meaning of most of the sentences in the last paragraph of the introduction is hard to interpret. It is important to check carefully for clarity.

b.      Section 2: The presentation of the methodology needs serious improvement. It is hard to understand the authors' explanations/descriptions.

                                                    i.     There are no citations for the statements in sections 2.1, 2.2 and 2.3.

                                                   ii.     The statement in the last paragraph of section 2.2 is weird.

·        “The detailed related characteristic pair establishment method will 157 not be given in this article. Those who are interested can contact the author for a discussion.”  

It will be hard to judge the work if your methodology is well presented. Scientific discoveries build on the open sharing of ideas in published studies.

c.      Section 3.2:

                                                    i.     Re-phrase this: “The yield strength is selected. The standard 185 value is 300MPa”

                                                   ii.     Provide references for the constitutive law in Table 3.

                                                  iii.     Figure 4 is not referenced within the text of the manuscript.

                                                  iv.     The explanation of the finite element methodology needs to be improved for clarity.

d.      Section 4:

                                                    i.     In Figure 5, what criterion is used to determine “a pronounced qualitative change at 969kN”?

                                                   ii.     What is the meaning of “cloud diagram”?

                                                  iii.     What does the PEEQ in the legend of Figure 6 indicate?

                                                  iv.     Explain what ELSED denote.

                                                   v.     Explain what the vertical axis of Figures 7 and 8 indicates.

                                                  vi.     You need to relate the observations from your study with previous work. The significance of the results in Figures 7 and 8 is not clear. How is this useful for the practical designs of concrete-filled steel tubes?

                                                vii.     There is very limited validation done to support the accuracy of the results obtained.

Round 2

Reviewer 2 Report

The authors have done a great job with the revised manuscript.